# Examining the Antecedents of Novice STEM Teachers’ Job Satisfaction: The Roles of Personality Traits, Perceived Social Support, and Work Engagement

**DOI:** 10.3390/bs14030214

**Published:** 2024-03-06

**Authors:** Zhaochun Wang, Haozhe Jiang, Wu Jin, Jin Jiang, Jiahong Liu, Jia Guan, Yangxi Liu, Enlin Bin

**Affiliations:** 1East China Normal University, Shanghai 200241, China; 18708886762@163.com (Z.W.); king_education@163.com (W.J.); ljh_star_three@163.com (J.L.); liuyangxiliu2014@163.com (Y.L.); 2Lijiang Teachers College, Lijiang 674199, China; 3The Fifth School Affiliated to East China Normal University, Shanghai 201808, China; 4Center for Educational Technology and Resource Development, Ministry of Education (National Center for Educational Technology, NCET), Beijing 100031, China; guan1820110@126.com; 5Teachers College for Vocational and Technical Education, Nanning Normal University, Nanning 530001, China; enly1829001@gmail.com

**Keywords:** job satisfaction, novice STEM teacher, personality trait, perceived social support, work engagement

## Abstract

It is reported that teachers’ satisfaction levels are lower and stress levels are higher than other professional groups in many countries. This is especially true for science, technology, engineering, and mathematics (STEM) teachers. Considering job satisfaction has a direct impact on turnover intention, low satisfaction levels have led to high turnover rates and thus exacerbated the shortages of STEM teachers. Hence, there is an urgent need to explore the antecedents of novice STEM teachers’ job satisfaction. This study proposes a new model to show how novice STEM teachers’ personality traits and perceived social support influence their job satisfaction. A total of 2592 novice STEM teachers in eighteen randomly selected high schools in China were recruited. Data were analyzed using the structural equation modeling approach. The results illustrated that most novice STEM teachers had low levels of job satisfaction. Novice STEM teachers’ personality traits could directly and indirectly impact their job satisfaction. Meanwhile, novice STEM teachers’ job satisfaction was also directly influenced by work engagement and indirectly influenced by their perceived social support. Our findings theoretically contribute to the understanding of the determinants of STEM teachers’ job satisfaction and have important practical implications.

## 1. Introduction

Research on job satisfaction (JS) has a long history [1]. Numerous studies have indicated that JS represents one key indicator of employees’ psychological well-being at work [2,3] and that it is one significant antecedent of employees’ work performance [4], work behavior [5], commitment [6], job stress [7], turnover intention [8], and organization success [9]. Notably, it is reported that teachers’ satisfaction levels are lower and stress levels are higher than other professional groups in many countries [10,11]. This is especially true for science, technology, engineering, and mathematics (STEM) teachers [12,13]. Considering that JS can directly impact turnover intention [8], low satisfaction levels have led to high turnover rates and thus exacerbated the shortages of STEM teachers [13]. Hence, to address this issue, it is very necessary to explore what factors influence STEM teachers’ JS. Furthermore, studies have revealed the significance of STEM teachers’ JS in the process of improving the quality of STEM education [14] and students’ STEM learning outcomes [15]. This also highlights the importance of increasing STEM teachers’ JS. However, to date, few studies have focused on the predictors of STEM teachers’ JS [16].

There is a consensus that the initial years of employment are one of the most important segments of teachers’ professional careers [17]. If teachers do not feel satisfied with their jobs in their novice stages, they may be dissatisfied with their jobs throughout their teaching careers [17]. Hence, more attention should be paid to the exploration of novice teachers’ JS and its antecedents. However, until now, very few studies have set out to understand novice STEM teachers’ JS.

Nowadays, the job demands–resources (JD-R) model is widely applied to explore teachers’ JS [18,19]. Despite this, limited studies have confirmed its validity in the field of STEM teacher education [16]. Furthermore, some scholars have argued that “personal resources can have the same function as job resources” and thus suggested that personal resources should also be incorporated in the model [20] (p. 2). However, this argument has not been widely substantiated.

Motivated by the abovementioned three gaps, our study explores the antecedents of novice STEM teachers’ JS. Prior research has found that teachers’ JS may be decided by many antecedents [21], such as work conditions [22], work–life balance [23], and so on. In light of the JD-R model, we focus on the roles of teachers’ personality traits (PTs) and perceived social support (PSS), which are representative factors of personal resources [24] and job resources [25], respectively, on JS. In addition, according to the JD-R model, we examine the mediating role of work engagement between the two factors and JS. A cross-sectional methodological approach is adopted, and 2592 novice STEM teachers from both urban and rural areas in China are recruited in this study. A new model is validated to show how novice STEM teachers’ personality traits, perceived social support, and work engagement influence their JS.

## 2. Literature Review and Hypotheses

The JD-R model was first proposed by [26]. It posits that job characteristics can be classified as job resources and job demands [26]. Job demands related to work cost negatively impact employee well-being, while job resources related to work support positively impact employee well-being [26]. It should be noted that the impact of job resources and job demands on employee well-being is independent [16]. In other words, when it comes to employee well-being (including JS), job resources and job demands can be discussed together or separately. We mainly examine the effects of job resources on JS in this study.

Specifically, based on [26], job resources are “those physical, social, or organizational aspects of the job that may do any of the following: (a) be functional in achieving work goals; (b) reduce job demands and the associated physiological and psychological costs; (c) stimulate personal growth and development” (p. 501). Particularly, social support represents an important kind of job resource. This is because sufficient support from employees’ families, colleagues, friends, and others in their social networks can help individuals achieve their work goals, improve their psychological states, and promote their professional development [27]. The term perceived social support (PSS) can be used to describe “the degree to which individuals believe they have social support” [28] (p. 63).

The significant effects of PSS on JS have also been confirmed among different kinds of employees, such as bank staff [29], traffic enforcement agents [30], and nurses [31]. However, this impact has yet to be validated among STEM teachers. We hypothesize the following:

**Hypothesis** **1 (H1).**
*Novice STEM teachers’ perceived social support will significantly impact their job satisfaction.*


Over the past years, much effort has been made to develop the JR-D model [16,32]. For instance, it is argued that individuals who have sufficient personal resources can also receive a series of work support, which may further increase employees’ satisfaction [20]. Specifically, personal resources describe an individual’s sense of their capabilities to successfully control and influence their environments [33]. Individuals without personal resources are more likely to be affected by the negative aspects of their working environments [20,32]. On the contrary, individuals with rich personal resources are more likely to be resilient and may not be influenced by the working environments easily [20,32]. In this sense, personal resources are no less important than job resources and should also be considered when predicting JS [20,32]. As personal resources are as important as job resources in predicting JS [20,32], it may be meaningful to incorporate personal resources into the JR-D model.

Particularly, numerous studies have pointed out that individuals’ personality traits (PTs) are their important personal resources [33,34,35,36]. Specifically, the term PT is defined as “the characteristics that are stable over time, provide the reasons for the person’s behavior, and are psychological in nature” [26] (p. 448). Generally, PTs include five big factors, namely, openness, conscientiousness, extroversion, agreeableness, and neuroticism/emotional stability [37,38]. Emotional stability is the antonym of neuroticism [38]. In our study, we choose to use the term emotional stability instead of neuroticism. PTs can, to a large extent, determine individuals’ “affective, behavioral, and cognitive style” [26] (p. 440), including the style of perceived social support. To date, the impact of PTs on PSS has been validated among high school students [39], university students [40], and middle-aged adults [41]. However, this impact has yet to be confirmed among STEM teachers. We hypothesize the following:

**Hypothesis** **2 (H2).**
*Novice STEM teachers’ personality traits will significantly impact their perceived social support.*


As aforementioned, it is argued that personal resources are also significant antecedents of JS and should be incorporated into the JR-D model [18,20]. A lot of research has also indicated that employees’ PTs, as one of the personal resources, will predict their satisfaction [37,42,43,44]. For instance, one study detected that the big five factors of PTs had a multiple correlation of 0.41 with JS [37]. This revealed that PTs, as a composite construct consisting of five factors, is one significant antecedent of JS [37]. However, the impact of PTs on JS has yet to be confirmed among STEM teachers. We hypothesize:

**Hypothesis** **3 (H3).**
*Novice STEM teachers’ personality traits will significantly impact their job satisfaction.*


Work engagement (WE) is a “positive, fulfilling, work-related state of mind that is characterized by vigor, dedication, and absorption” [45] (p. 74). The JD-R model postulates that WE can mediate the effects of job resources on employees’ psychological well-being (including JS) [20,45,46]. Karatape et al. also proposed a conceptual model illustrating that sufficient job and personal resources can impact WE directly, and WE may further impact JS [47]. On the one hand, job and personal resources will help employees realize their goals, and employees with sufficient job and personal resources are more likely to be committed and engaged as they can derive fulfillment from their work [48]. On the other hand, engaged teachers usually perform well in their schools and gain positive experiences [49,50] and thus will feel satisfied with their jobs [50]. As TS is one important factor of personal resources and PSS is one important factor of job resources, it can be assumed that WE may mediate the impact of TS and PSS on JS. Specifically, firstly, prior research has elucidated that the big five factors of PTs can exert effects on WE [51]. For instance, in one meta-analysis, researchers confirmed that all of the five PTs are significant antecedents of WE [51]. Secondly, the assumption that PSS can predict WE has also been justified in two previous studies [52,53]. Thirdly, the assumption that WE is one antecedent of JS has also been verified in one prior meta-analysis [54]. Notably, although these relationships have been confirmed in some prior studies, they have seldom been detected among novice STEM teachers. We hypothesize the following:

**Hypothesis** **4 (H4).**
*Novice STEM teachers’ perceived social support will significantly impact their work engagement.*


**Hypothesis** **5 (H5).**
*Novice STEM teachers’ personality traits will significantly impact their work engagement.*


**Hypothesis** **6 (H6).**
*Novice STEM teachers’ work engagement will significantly impact their job satisfaction.*


**Hypothesis** **7 (H7).**
*Novice STEM teachers’ work engagement will mediate the impact of perceived social support on job satisfaction.*


**Hypothesis** **8 (H8).**
*Novice STEM teachers’ work engagement will mediate the impact of personality traits on job satisfaction.*


Additionally, considering the hypothesized impact of PTs on PSS (see H2) and the hypothesized mediating role of WE between PSS and JS (see H7), it can be assumed that PSS and WE may serially mediate the impact of PTs on JS. However, to the best of our knowledge, this serial mediating effect has never been validated before. Hence, we propose our ninth hypothesis to help better understand the roles of PSS and WE in predicting JS.

**Hypothesis** **9 (H9).**
*The impact of novice STEM teachers’ personality traits on job satisfaction will be serially mediated by perceived social support and work engagement.*


Finally, considering the hypothesized impact of PTs on PSS (see H2) and the hypothesized impact of PSS on JS (see H1), we propose our final hypothesis. It should be noted that this mediating effect has never been verified in STEM teacher education.

**Hypothesis** **10 (H10).**
*Novice STEM teachers’ perceived social support will mediate the impact of personality traits on job satisfaction.*


Based on the above-mentioned arguments, we propose the conceptual model for our study (see Figure 1). Our model is drawn from the JD-R model [26]. Firstly, the JD-R model suggests PSS (as one kind of job resource) will impact WE and JS [26]. Secondly, we adopt Kwon and Kim’s and Radic et al.’s advice and incorporate personal resources into the JD-R model [20,46]. In our model, personal resources are given the same importance as job resources. Hence, the variable PTs (as one kind of personal resource) is assumed to influence WE and JS. Thirdly, considering the impact of WE on JS indicated by the JD-R model [20,45,46], our model also assumes that WE can mediate the impact of PSS and PTs on JS.

The proposed conceptual model has filled at least three theoretical gaps. Firstly, although the JD-R model’s validity in predicting employees’ JS has been justified in various fields (e.g., healthcare professionals, mariners) [18,19], to date, it has rarely been applied in STEM education, and its power in predicting STEM teachers’ JS remains unknown [16]. To fill this theoretical gap, we propose the conceptual model, which is based on the JD-R model, to explore the antecedents of novice STEM teachers’ JS. Secondly, prior studies applying the JD-R model paid little attention to the impact of personal resources on JS [20,32]. In other words, the roles of personal resources were usually ignored in studies applying the JD-R model. To fill this gap, the proposed conceptual model expands the JD-R model by incorporating personal resources. It articulates that personal resources are as important as job resources in predicting JS. Thirdly, most prior studies paid attention to the direct relationships among the JD-R model [16,17], while the indirect relationships were seldom examined. To fill this gap, our model posits that PSS and WE can serially mediate the impact of PTs on JS, which has never been studied before. This mediating effect adds to the current understanding of the roles of PSS and WE in predicting JS.

## 3. Method

As part of a larger project, which aimed to enhance STEM teachers’ well-being and facilitate their professional development [14], our research goal was to examine the relationships among novice STEM teachers’ PTs, PSS, WE, and JS. Thus, a quantitative cross-sectional survey was the most efficient and appropriate method [55], as it can help collect data regarding these variables in a relatively short period of time [56,57].

### 3.1. Participants

This study was conducted in eighteen randomly selected high schools in China. Among them, nine were located in the east of China and nine were in the west of China. Notably, twelve of them were urban schools, and two of them were private schools. A development gap exists between eastern and western Chinese regions, and this further leads to a gap in educational development between eastern and western Chinese schools [58]. For instance, teachers in the east of China are more likely to earn higher salaries than those in the west, and eastern Chinese schools are more likely to have more advanced facilities (e.g., virtual reality devices) for STEM education than western Chinese schools [14,58]. In this sense, STEM teachers both in developed regions and developing regions were included in our sample. Almost all types of high schools in China (i.e., public/private schools, urban/rural schools, and schools located in developed/developing regions) were included in our study. Hence, our sample is, to some degree, representative.

A total of 2690 teachers from the eighteen schools voluntarily filled in our questionnaires. All of them were novice STEM teachers who had less than two years of teaching experience. During our initial check, we removed 98 incomplete or invalid responses, and the effective rate was 96.36%. Among the 2592 effective responses, 1469 (56.67%) participants were male STEM teachers, and 1123 (43.33%) participants were female STEM teachers. A total of 1697 (65.47%) participants were from Eastern Chinese regions and 895 (34.53%) participants were from western Chinese regions. A total of 1092 (42.13%) participants were science teachers, 573 (22.10%) participants were technology teachers, and 927 (35.76%) participants were mathematics teachers. In China, engineering is not taught in high schools, and thus no engineering teachers were included in our sample. A total of 1339 (51.66%) participants had the experience of implementing integrated STEM teaching. Particularly, 2123 (81.91%) participants once attended courses related to STEM teaching during their undergraduate or postgraduate studies.

### 3.2. Instrument Development

Our study involved four instruments, which were respectively used to measure novice STEM teachers’ PTs, PSS, WE, and JS. Specifically, the scale for measuring personality traits was adapted from Rammstedt and John [59], the scale for measuring perceived social support was adapted from Zimet et al. [60], the scale for measuring work engagement was adapted from Schaufeli et al. [61], and the scale for measuring job satisfaction was adapted from Ho and Au [62]. Particularly, scales for measuring PTs are usually very long, and participants cannot complete them in a limited time [59]. Given this, Rammstedt and John developed a scale including only 10 items, which can be completed in two minutes [59]. This short instrument also has acceptable validity and reliability [59].

The procedure of instrument development was based on Jiang et al.’s guidelines [63]. In the first step, these original scales were revised and adapted so that they could target STEM teachers. In the second step, we developed the Chinese scales. Specifically, two language experts who were familiar with psychology areas translated the English scales into Chinese. Five experts in psychology who were skilled in English then checked and improved the translation. Next, two other language experts translated the Chinese scales into English. Five experts confirmed that there were hardly any differences between the translations and the original ones. In the third step, following Jiang et al.’s suggestion [63], we conducted a pilot test, in which 367 novice STEM teachers were involved. After the pilot test, we analyzed the data using AMOS version 21. Two items (i.e., PT10 and JS5) were removed as their factor loadings were lower than 0.45 [64]. After removing the two items with low factor loadings, the remaining items were included in the final instruments, which were used in our formal data collection. The items of our final instruments are also described in Appendix A. We examined the reliability and validity of our instruments using the confirmatory factor analysis (CFA) technique and by testing the measurement model. The results regarding the instruments’ reliability and validity are presented in the Section 4.

### 3.3. Data Collection and Data Analysis

Ethical endorsement for our study was obtained. During the data collection processes, we strictly followed ethical principles. In the formal data collection, the teaching administrative departments of the eighteen randomly selected schools helped us invite novice STEM teachers to participate in our study. Novice STEM teachers who expressed interest in taking part in our research were asked to complete the survey questionnaires anonymously. It took around 15 min to complete the questionnaire. On the one hand, participants were allowed to quit at any time. On the other hand, participants were told that the answers would be used in research only, and their personal information was also strictly protected.

Adopting the structural equation modeling approach [65], we used software named AMOS version 21 to conduct the data analysis. We examined the validity and reliability of our instruments by using the CFA technique and testing the measurement model. We then examined our hypotheses by testing the structural model. As for H7–H10, bootstrapping was also used as a resampling method to examine the mediating effects. Gender and age were treated as control variables in the analysis.

Based on Hu and Bentler’s suggestion [66], several goodness-of-fit indices were utilized in our data analysis, namely, chi-square (χ^2^), degrees of freedom (df), the comparative fit index (CFI), the Tucker–Lewis index, and the root mean square error of approximation (RMSEA). Generally, it is recommended that a model is considered to have a good fit with the data when CFI ≥ 0.9, TLI ≥ 0.9, and RMSEA ≤ 0.08 [66].

## 4. Results

In this section, we first present the results of the CFA technique and the measurement model to indicate the validity and reliability of our instruments. We then present the results of the structural model to scrutinize our hypotheses.

### 4.1. The Results of the Measurement Model

According to the AMOS output, the measurement model with four constructs (i.e., PTs, PSS, WE, and JS) fitted the data well with CFI  =  0.908, TLI  =  0.900, RMSEA = 0.076, and SRMR = 0. 042.

To confirm the construct validity, we calculated the composite reliability (CR) for each construct [64]. As Table 1 shows, all the CR values were above 0.7 [64]. Therefore, our instruments possessed good construct validity [64].

To confirm the convergent validity, we calculated the average variance extracted (AVE) for each construct [64]. As Table 1 shows, all the AVE values were above 0.5 [64]. Therefore, our instruments possessed good convergent validity [64].

Notably, our results showed that most novice STEM teachers had low levels of job satisfaction as the mean values for all the items of job satisfaction were below 4.0. This indicated that most novice STEM teachers were not satisfied with their current jobs. Moreover, the mean value for JS4 was below 3.0, which indicated that a large percentage of teachers would like to change their jobs.

We followed Fornell and Larcker’s recommendations to confirm the discriminant validity [67]. The results are shown in Table 2. For each construct, the value of the square roots of its AVE was larger than its correlations with other constructs. Therefore, our instruments possessed good discriminant validity [67].

The maximum factor variance explained was 30.417%. As it was smaller than 40%, significant common method bias did not exist [64].

In summary, by using the CFA and testing the measurement model, the validity and reliability of our instruments were justified.

### 4.2. The Results of the Structural Model

The structural model also fitted the data well with CFI  =  0.908, TLI  =  0.900, RMSEA = 0.076, and SRMR = 0.042. Figure 2 and Table 3 show the results of hypotheses testing (H1–H6). Novice STEM teachers’ PTs directly impacted their PSS (β = 0.633, *p* < 0.001). As such, H2 was accepted. Novice STEM teachers’ WE was directly impacted by PSS (β = 0.335, *p* < 0.001) and PTs (β = 0.524, *p* < 0.001). Furthermore, novice STEM teachers’ JS was directly impacted by PTs (β = 0.254, *p* < 0.001) and WE (β = 0.316, *p* < 0.001). However, the impact of novice STEM teachers’ PSS on their JS was not significant (β = −0.044, *p* = 0.115). Therefore, H3 and H6 could be accepted, while H1 should be rejected.

Table 4 shows the results of hypotheses testing (H7–H10), where mediating effects were involved. WE significantly mediated the impact of PSS on JS (β = 0.106, *p* < 0.001) and the impact of PTs on JS (β = 0.166, *p* < 0.001). As such, H7 and H8 could be accepted. PSS and WE serially mediated the impact of PTs on JS. Therefore, H9 was also confirmed. However, the indirect impact of PTs on JS via PSS was not significant (β = −0.027, *p* = 0.178). Therefore, H10 was also rejected.

Our model showed that PTs and PSS could impact JS directly and indirectly. Specifically, there was one indirect path between PSS and JS (i.e., PSS→WE→JS) and three indirect paths between PTs and JS (i.e., PTs→WE→JS, PTs→PSS→WE→JS, and PTs→PSS→JS). Particularly, one indirect path (i.e., PTs→PSS→JS) was not significant, while the other three indirect paths were significant. Table 5 summarizes the direct, indirect, and total effects of PSS, PTs, and WE on JS. Among the three determinants of STEM teachers’ JS, the construct of PTs was the most powerful one (β = 0.459, *p* < 0.001). Among the effects of PTs on JS, 55.34% were direct effects, while 44.66% were indirect effects. The direct effects of PSS on JS (β = −0.044, *p* > 0.05) were not significant, while the indirect (β = 0.106, *p* < 0.001) and total (β = 0.062, *p* < 0.05) effects were significant.

## 5. Discussion

### 5.1. Theoretical Contribution

STEM education has been advocated and intensively implemented all over the world over the past years [12,57,63,68]. However, it is reported that there is a severe shortage of qualified STEM teachers around the world [69,70,71], which has hindered the implementation of STEM education reforms [12,69,70]. One of the most important causes of the shortage is STEM teachers’ low levels of JS and high turnover rates [13]. Hence, there is an urgent need to explore the determinants of STEM teachers’ JS [16]. This study theoretically contributed to the limited understanding of the determinants of STEM teachers’ JS in the following aspects.

First of all, in line with [18], our results indicated that most STEM teachers had low levels of JS and a large percentage of teachers would like to change their jobs. Notably, our participants were novice STEM teachers, which has seldom been specifically investigated before. As aforementioned, they are very likely to feel dissatisfied with their jobs throughout their teaching careers if they have low levels of job satisfaction in their novice stages [17]. This phenomenon is worrisome as it might exacerbate the shortages of STEM teachers around the world. Based on this finding, our study called for more theoretical studies on the determinants of novice STEM teachers’ JS and highlighted the urgent need to enhance their JS.

Second, contrary to what the JD-R model posits [26] and prior findings [29,30,31], our results revealed that the direct impact of PSS, as one kind of job resource, on JS was not significant among novice STEM teachers. This might be because STEM teaching is professional and stressful [12], and social support is not powerful enough to help novice STEM teachers relieve job anxiety, improve job efficiency, and enhance job satisfaction directly.

Third, our study empirically supported Kwon and Kim’s and Radic et al.’s suggestions that personal resources are no less important than job resources and should also be considered when predicting JS [20,32]. Our results showed that novice STEM teachers’ PT, as one of the most important personal resources, directly and significantly impacted their JS. By incorporating one kind of personal resource in our model, our findings theoretically advance the understanding of the determinants of JS and expand the JD-R model. Hence, when considering the antecedents of JS, personal resources such as PTs, which are ignored by the JD-R model, should also be paid attention to.

Fourth, our findings illustrated the relationship between personal resources and job resources by confirming the impact of PTs on PSS. To date, this impact has been validated among high school students [39] university students [40], and middle-aged adults [41]. This study validated this impact in a new context, namely, STEM teacher education. It should be noted that personal resources are not included in the original JD-R model [26]. As such, our study proposed a novel topic regarding the relationships between personal resources and other factors (e.g., job resources) of the JD-R model.

Fifth, our study also articulated that WE played the mediating role between resources and employees’ psychological well-being (including JS). Although the mediating role of WE has been reported in prior studies [20,45,46], we confirmed it in a new context, namely, STEM teacher education. Our study further illustrated that PSS (as one kind of job resource) and PTs (as one kind of personal resource) could be indirectly related to JS through WE. Notably, the significant indirect relationship between PTs and JS (i.e., PTs→WE→JS, PTs→PSS→WE→JS) has seldom been revealed before. Meanwhile, although the direct association between PSS and JS among novice STEM teachers was not significant, the indirect effects were significant. In this sense, although PSS could not directly shape novice STEM teachers’ JS significantly, our study still indicated that it should be considered an important variable when exploring the factors contributing to STEM teachers’ JS due to its significant indirect effects.

### 5.2. Practical Implications

Considering the urgent need to enhance novice STEM teachers’ JS, our study has important practical implications. Based on our model, novice STEM teachers’ JS can be promoted by modifying their PTs partly and slightly, enhancing their PSS, and improving WE.

First, although PTs are relatively stable characteristics, research has also shown that part of them can be modified slightly through professional training [44]. For instance, individuals’ emotional stability, one of the most important trait domains, can be fostered through intervention [44]. As such, more teacher professional programs can be provided, where STEM teachers can learn how to maintain emotional stability and relieve stress. Specifically, to enhance novice STEM teachers’ emotional stability, it is suggested that novice STEM teachers should learn some strategies to regulate their emotions (e.g., attentional deployment) [12]. Novice STEM teachers should know how to use the strategy of attentional deployment so as to pay attention to the positive aspects of their working environments and overcome negative emotions in their teaching [12]. Meanwhile, as novice STEM teachers do not have enough teaching experience, it is common for them to encounter unsuccessful teaching [12]. Hence, novice STEM teachers should learn to appropriately treat their unsuccessful teaching. They should know how to learn from their unsuccessful teaching instead of just feeling depressed [12]. In addition, novice STEM teachers should learn to notice and appreciate their students’ improvement, which will also help them become optimistic [72].

Secondly, more social support, especially organizational support [73], should be provided for novice STEM teachers. For instance, schools may contemplate implementing sessions for sharing and debriefing, creating an environment where teachers are motivated to openly discuss their teaching encounters to nurture supportive connections among peers. New teachers might benefit from mentoring sessions to enhance their ability to articulate challenging experiences at work. At the organizational level, human resources managers could encourage a supportive internal workplace by implementing a suitable code of conduct and arranging meetings to promote and cultivate a civil culture of learning and collaboration. In recent years, many countries have released a lot of policies to support STEM teaching [12,57,63,68,74,75]. Schools should help novice STEM teachers realize these policies so that novice STEM teachers can perceive that what they are doing is supported by the governments and societies [12].

Finally, WE is not only a direct determinant of novice STEM teachers’ JS but also plays an important mediating role in the impact of personal and job resources on JS. Hence, it is very important to improve novice STEM teachers’ WE. For instance, most novice STEM teachers do not have enough work skills [12], and this will further decrease their work engagement [76]. Enhancing work skills will help employees improve their work engagement [76,77]. Therefore, it is essential to help novice STEM teachers improve their work skills (e.g., classroom teaching skills, classroom management skills, etc.) through teacher professional training. For another instance, stronger teaching motivation will lead to higher levels of work engagement [77]. Schools can increase novice STEM teachers’ teaching motivation by several measures, such as creating a supportive atmosphere, helping them realize their goals of professional development [77], and reducing their emotional exhaustion [12,78].

### 5.3. Limitations and Future Research

Our study has some limitations. Due to the cross-sectional design of our research, it is not possible to infer causal relationships. Furthermore, this study is solely dependent on self-reported measures, thereby being subject to the limitations inherent in such a methodology. Future research should consider employing longitudinal techniques to capture changes over time and demonstrate causal linkages and consider integrating data from various sources of information to enhance the depth and reliability of findings. Although the data were collected from a large sample of novice teachers from eighteen high schools, the data collection was limited to the Chinese context. Future cross-cultural replications should be conducted to verify whether our results are generalizable to other countries, including comparisons between novice and senior teachers. Another limitation arises from the inability to eliminate selection bias due to the voluntary participation of respondents in this research. Subsequent research endeavors might consider incorporating incentives to motivate participants, aiming to ensure the inclusion of all individuals affiliated with a specific emergency organization and thereby mitigating potential bias. Furthermore, the current findings are limited by the exclusion of variables relevant to comprehending how and when novice STEM teachers are satisfied with their jobs. However, it is impossible to include a lot of variables in a single structural equation model [55]. Future studies are therefore needed to expand our model by incorporating additional theories or frameworks to provide a more in-depth understanding of the intricate interactions among various factors influencing JS. In addition, it is recommended that future studies explore the relationships between teachers’ demographic variables (e.g., learning experiences, educational background) and their JS. Finally, the COVID-19 pandemic dramatically impacted STEM education [71,72], and STEM teachers were required to implement online teaching, which made some STEM teachers feel uneasy [72]. Hence, it is meaningful for future studies to explore the impact of the COVID-19 pandemic and technology-enabled teaching on STEM teachers’ JS. Meanwhile, as aforementioned, many countries have released a lot of policies to support STEM teaching [12,57,63,68,74,75], and future studies can also explore the impact of these new policies on teachers’ JS.

## 6. Conclusions

Job satisfaction is a key predictor of employees’ psychological well-being [2,3]. This study proposes a new model showing how novice STEM teachers’ personality traits and perceived social support influence their job satisfaction. Our results indicated that most novice STEM teachers had low levels of job satisfaction. Novice STEM teachers’ personality traits could directly and indirectly impact their job satisfaction. Meanwhile, novice STEM teachers’ job satisfaction was also directly influenced by work engagement and indirectly influenced by their perceived social support. This study theoretically contributes to the understanding of the determinants of STEM teachers’ job satisfaction and has important practical implications.

## Figures and Tables

**Figure 1 behavsci-14-00214-f001:**
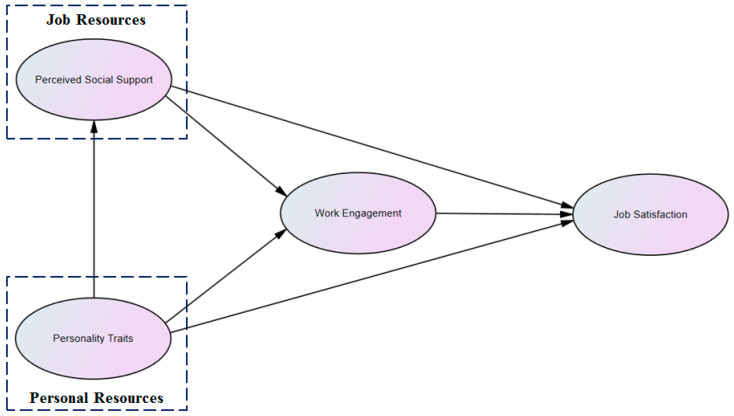
The conceptual model.

**Figure 2 behavsci-14-00214-f002:**
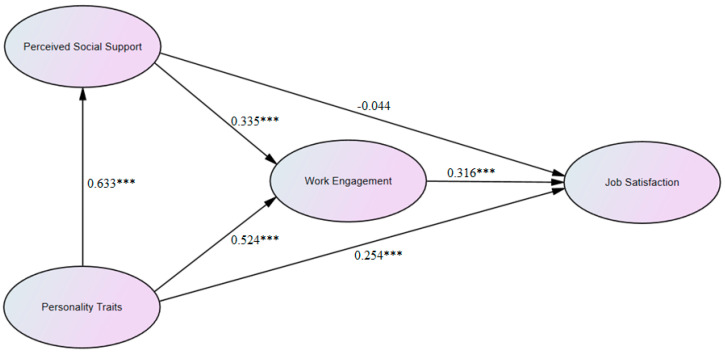
The results of hypotheses testing (H1–H6), *** *p* < 0.001.

**Table 1 behavsci-14-00214-t001:** The CFA results.

Construct	Item	M	SD	Loading	CR	AVE
**Personality traits**	PT1	4.78	1.57	0.833	0.965	0.757
	PT2	4.33	1.65	0.749		
	PT3	4.58	1.62	0.818		
	PT4	4.43	1.64	0.774		
	PT5	4.76	1.53	0.905		
	PT6	4.73	1.53	0.960		
	PT7	4.74	1.51	0.963		
	PT8	4.78	1.53	0.960		
	PT9	4.38	1.54	0.834		
**Perceived social support**	PSS1	4.81	1.49	0.817	0.898	0.529
	PSS2	4.79	1.38	0.863		
	PSS3	4.33	1.39	0.703		
	PSS4	4.40	1.41	0.692		
	PSS5	4.10	1.28	0.628		
	PSS6	3.41	1.47	0.812		
	PSS7	5.07	1.33	0.696		
	PSS8	4.83	1.35	0.555		
**Work engagement**	WE1	4.79	1.24	0.884	0.976	0.722
	WE2	4.84	1.31	0.898		
	WE3	4.93	1.30	0.890		
	WE4	4.75	1.30	0.882		
	WE5	4.71	1.28	0.850		
	WE6	4.63	1.44	0.727		
	WE7	4.66	1.38	0.724		
	WE8	5.15	1.37	0.841		
	WE9	4.84	1.33	0.871		
	WE10	4.80	1.40	0.905		
	WE11	4.63	1.44	0.755		
	WE12	4.55	1.28	0.918		
	WE13	4.54	1.31	0.955		
	WE14	4.62	1.30	0.923		
	WE15	4.68	1.33	0.892		
	WE16	4.26	1.58	0.603		
**Job satisfaction**	JS1	3.31	1.55	0.810	0.918	0.737
	JS2	3.11	1.66	0.894		
	JS3	3.27	1.72	0.837		
	JS4	2.95	1.62	0.890		

M: mean, SD: standard deviation, CR: the composite reliability, AVE: the average variance extracted. All items are described in Appendix A.

**Table 2 behavsci-14-00214-t002:** The results of the discriminant validity.

Construct	PT	PSS	WE	JS
**PT**	0.870			
**PSS**	0.633 ***	0.727		
**WE**	0.737 ***	0.667 ***	0.850	
**JS**	0.459 ***	0.328 ***	0.474 ***	0.859

***: *p* < 0.001. The bold numbers refer to the square roots of the AVE values.

**Table 3 behavsci-14-00214-t003:** The results of hypotheses testing (H1–H6).

Hypothesis	Path	β	SE	C.R.	Result
**H1**	PSS→JS	−0.044	0.036	−1.556	Reject
**H2**	PT→PSS	0.633 ***	0.017	28.357	Accept
**H3**	PT→JS	0.254 ***	0.029	8.517	Accept
**H4**	PSS→WE	0.335 ***	0.019	15.326	Accept
**H5**	PT→WE	0.524 ***	0.015	22.874	Accept
**H6**	WE→JS	0.316 ***	0.048	9.544	Accept

β: standardized coefficient, ***: *p* < 0.001.

**Table 4 behavsci-14-00214-t004:** The results of hypotheses testing (H7–H10).

Hypothesis	Path	β	95% CI	Result
**H7**	PSS→WE→JS	0.106 ***	[0.098, 0.183]	Accept
**H8**	PT→WE→JS	0.166 ***	[0.122, 0.199]	Accept
**H9**	PT→PSS→WE→JS	0.067 ***	[0.046, 0.087]	Accept
**H10**	PT→PSS→JS	−0.027	[−0.063, 0.012]	Reject

95% CI: 95% bias-corrected percentile confidence intervals, ***: *p* < 0.001.

**Table 5 behavsci-14-00214-t005:** The direct, indirect, and total effects of PSS, PTs, and WE on JS.

Construct	PSS	PT	WE
**Direct effects**	−0.044	0.254 ***	0.316 ***
**Indirect effects**	0.106 ***	0.205 ***	/
**Total effects**	0.062 *	0.459 ***	0.316 ***

*: *p* < 0.05, ***: *p* < 0.001.

## Data Availability

The data are available upon request from researchers who meet our eligibility criteria. Kindly contact the corresponding authors privately by e-mail.

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
