# Peer review of "Examining the Antecedents of Novice STEM Teachers’ Job Satisfaction: The Roles of Personality Traits, Perceived Social Support, and Work Engagement"

_behavsci, 2024, doi:10.3390/bs14030214_

Round 1

Reviewer 1 Report

Comments and Suggestions for Authors

The conceptual framework of the present research is well explained, as well as the underlying research questions. The quantitative methodology is appropriate given the large number of participants. On the other hand, the description of the population would have been relevant if we knew more about their scientific training in the case of each of the three groups of teachers (science, technology, and mathematics). It would have been relevant to specify in the section the limits of the results put forward to see if a relation is established between teachers' scientific and didactic training and their attitude concerning, for example, their commitment and satisfaction in their profession.

Author Response

Reviewer #1: The conceptual framework of the present research is well explained, as well as the underlying research questions. The quantitative methodology is appropriate given the large number of participants. On the other hand, the description of the population would have been relevant if we knew more about their scientific training in the case of each of the three groups of teachers (science, technology, and mathematics). It would have been relevant to specify in the section the limits of the results put forward to see if a relation is established between teachers' scientific and didactic training and their attitude concerning, for example, their commitment and satisfaction in their profession.

Authors’ responses: Thank you so much for your favorable comments. We have completely followed your suggestion and added the information in the subsection entitled “3.1 Participants”. We have also pointed out that it is very meaningful to explore the relationship between participants’ demographic variables and their job satisfaction in future studies in the subsection entitled “5.3 Limitations and future research”. The revisions have been highlighted in red. Thank you again for your helpful suggestions.

Reviewer 2 Report

Comments and Suggestions for Authors

Include a brief mention of the methodological approach (e.g., survey, longitudinal, cross-sectional) and the nature of the sample (e.g., novice STEM teachers from urban/rural areas, public/private schools) in the context. This would provide readers with a clearer context of the study.

In the introduction, clarify why the study focuses specifically on novice STEM teachers as opposed to experienced teachers or teachers from other disciplines. When mentioning the JD-R model, a brief explanation of the model would be beneficial for readers not familiar with it. Consider expanding upon why personal resources are hypothesized to be as important as job resources in this context.

The literature review is thorough, but it could be improved by providing a more critical analysis of the existing literature, perhaps noting any limitations or gaps that this study intends to fill. It's mentioned that the JD-R model is widely applied but has limited validation in STEM education—this is an important point that could be expanded upon to demonstrate the contribution of the present study. The hypotheses are clearly stated. However, it would be helpful to frame them within the context of the current literature, discussing how they align with or diverge from previous research findings.

The conceptual model combines JD-R with personality traits, but the rationale for bringing these two frameworks together could be explained more clearly. The measures used seem quite brief with only 3-4 items per construct. Justification for the reliability and validity of such brief measures should be provided.

The cross-sectional design means causality cannot be inferred, which should be highlighted in the limitations. The practical implications could be developed further with more specific, actionable suggestions beyond just "improve personality traits, social support, and work engagement."

The sample description should note how representatives the schools are of broader populations.

Definitions of key terms like job resources, personal resources, and work engagement would help readers unfamiliar with JD-R.

 References need to be formatted consistently throughout

Comments on the Quality of English Language

Editing for clarity and conciseness

Author Response

Reviewer #2: Include a brief mention of the methodological approach (e.g., survey, longitudinal, cross-sectional) and the nature of the sample (e.g., novice STEM teachers from urban/rural areas, public/private schools) in the context. This would provide readers with a clearer context of the study.

Authors’ responses: Thank you so much for this comment. We have followed your suggestion and included a brief mention of the methodological approach (e.g., survey, longitudinal, cross-sectional) and the nature of the sample (e.g., novice STEM teachers from urban/rural areas, public/private schools) in the Introduction section. The revisions have been highlighted in red.

Reviewer #2: In the introduction, clarify why the study focuses specifically on novice STEM teachers as opposed to experienced teachers or teachers from other disciplines. When mentioning the JD-R model, a brief explanation of the model would be beneficial for readers not familiar with it. Consider expanding upon why personal resources are hypothesized to be as important as job resources in this context.

Authors’ responses: Thank you for this comment.

Firstly, we have highlighted the reasons why our study focuses specifically on novice STEM teachers instead of experienced teachers in the Introduction section in red.

Secondly, we have highlighted the brief explanation of the JD-R model in the section entitled “2. Literature Review and Hypotheses” in red.

Thirdly, we have followed your kind suggestion, and expanded the reasons why personal resources are hypothesized to be as important as job resources in this context in the section entitled “2. Literature Review and Hypotheses”. The added sentences have been highlighted in red.

We agree with you that this information is very important. Meanwhile, we think a brief and straightforward Introduction section is important. Hence, we have carefully followed your suggestions and included the information in the section entitled “2. Literature Review and Hypotheses”.

Thank you again for your insightful suggestions.

Reviewer #2: The literature review is thorough, but it could be improved by providing a more critical analysis of the existing literature, perhaps noting any limitations or gaps that this study intends to fill. It's mentioned that the JD-R model is widely applied but has limited validation in STEM education—this is an important point that could be expanded upon to demonstrate the contribution of the present study. The hypotheses are clearly stated. However, it would be helpful to frame them within the context of the current literature, discussing how they align with or diverge from previous research findings.

Authors’ responses: Thank you so much for your comment. We have completely followed your suggestions and improved the literature review. Notably, we have added one paragraph to clarify that we have filled at least theoretical gaps (please see the last paragraph of the section entitled “2. Literature Review and Hypotheses”). The revisions have been highlighted in red.

Reviewer #2: The conceptual model combines JD-R with personality traits, but the rationale for bringing these two frameworks together could be explained more clearly. The measures used seem quite brief with only 3-4 items per construct. Justification for the reliability and validity of such brief measures should be provided.

Authors’ responses: Thank you so much for your comment.

Firstly, we have followed your advice and explained the rationale for bringing these two frameworks together more clearly in the section entitled “2. Literature Review and Hypotheses”. The revisions have been highlighted in red.

Secondly, JS has 4 items, PT has 9 items, PSS has 8 items, and WE has 16 items (for more details, please see the Appendix). Hence, only one measure is brief. The reliability and validity of our instruments have been reported in the subsection entitled “4.1 The Results of the Measurement Model”.

Reviewer #2: The cross-sectional design means causality cannot be inferred, which should be highlighted in the limitations. The practical implications could be developed further with more specific, actionable suggestions beyond just "improve personality traits, social support, and work engagement."

Authors’ responses: Thank you for this comment.

Firstly, we have followed your advice and highlighted the methodology limitations.

Secondly, we have followed your advice and included more details in the subsection entitled “5.2 Practical Implications”.

The revisions have been highlighted in red.

Reviewer #2: The sample description should note how representatives the schools are of broader populations.

Authors’ responses: Thank you so much for your comment. We have followed your advice and noted how representatives the schools are of broader populations. The revisions have been highlighted in red.

Reviewer #2: Definitions of key terms like job resources, personal resources, and work engagement would help readers unfamiliar with JD-R.

Authors’ responses: Thank you for this comment. We have highlighted the definitions of key terms like job resources, personal resources and work engagement in red.

Reviewer #2: References need to be formatted consistently throughout.

Authors’ responses: Thank you for your comment. We have formatted the references need throughout. Some references do not have volume, issue and page numbers as they are advance online publications.

Reviewer 3 Report

Comments and Suggestions for Authors

Review of the Literature
The submission's literature review successfully situates the study inside the Job Demands-Resources (JD-R) model's theoretical framework, illuminating a thorough comprehension of job demands, job resources, and their effects on worker well-being. In the context of new STEM instructors, the review emphasises the significance of perceived social support (PSS) as a crucial employment resource. The literature analysis effectively establishes the context for the hypothesis that PSS has a large impact on job satisfaction among this group by combining findings from multiple research.

But the literature evaluation may be improved by giving a more thoughtful analysis of conflicting results or gaps in the field's knowledge, especially in light of the particular difficulties that new STEM instructors encounter. Furthermore, adding research from various educational and geographic contexts could improve the review and provide a more worldwide viewpoint.

Method
The methodology section describes a methodical and transparent way to investigate the connections between the rookie STEM teachers' personality traits (PT), perceived social support (PSS), work engagement (WE), and job satisfaction (JS). It makes sense to utilise a quantitative cross-sectional survey, and structural equation modelling (SEM) is a suitable method for examining the intricate links suggested by the study's hypotheses. The study's dedication to ethical research procedures is demonstrated by the thorough explanation of the data collection procedure, which takes ethical considerations and the participation of instructional administration departments into account. Furthermore, a methodologically sound approach is demonstrated by the inclusion of control variables like age and gender and the use of AMOS software for SEM. However, there is still room for improvement. Although effective, the cross-sectional design restricts the capacity to establish causation between the variables under investigation. Longitudinal designs may prove useful in future studies to better capture changes over time and more clearly demonstrate causal linkages.Self-reported metrics run the risk of introducing bias. Further research with objective measures or third-party evaluations may improve the validity of the results.
-A more thorough description of the pilot testing procedure, including the justification for item removal and how the pilot test influenced the final survey instrument, would be beneficial to include in the technique section.

Additional information
- Theoretical Contribution: Although the study adds to our knowledge of novice STEM teachers' job satisfaction, it might incorporate additional theories or frameworks to enhance the analysis beyond the JD-R model. Examining other ideas might provide more in-depth understanding of the intricate interactions between various elements influencing work satisfaction.
- Practical Implications: While useful, the practice recommendations are a little too general. More targeted, doable solutions that are adapted to the particular difficulties faced by the educational sector will improve the piece. Giving specific instances of effective programmes or interventions could make these recommendations more useful.
- Limitations and Future Research: It is appropriate to acknowledge the limitations of this study, including its cross-sectional design and reliance on self-reported data. The conversation could be broadened, though, in order to address potential biases in greater detail and offer particular methodological enhancements, such mixed-methods or longitudinal designs, for upcoming research. Innovativeness: Although the study offers insightful conclusions, it could be more innovative to investigate lesser-known or recently discovered factors that influence job satisfaction, such as the effects of global crises like the COVID-19 pandemic, policy changes in education, or technological developments in the classroom.

Author Response

Reviewer #3: Review of the Literature
The submission’s literature review successfully situates the study inside the Job Demands-Resources (JD-R) model's theoretical framework, illuminating a thorough comprehension of job demands, job resources, and their effects on worker well-being. In the context of new STEM instructors, the review emphasises the significance of perceived social support (PSS) as a crucial employment resource. The literature analysis effectively establishes the context for the hypothesis that PSS has a large impact on job satisfaction among this group by combining findings from multiple research.

Authors’ responses: Thank you so much for your favorable comment.

Reviewer #3: But the literature evaluation may be improved by giving a more thoughtful analysis of conflicting results or gaps in the field's knowledge, especially in light of the particular difficulties that new STEM instructors encounter. Furthermore, adding research from various educational and geographic contexts could improve the review and provide a more worldwide viewpoint.

Authors’ responses: Thank you so much for your comment. We have completely followed your suggestions and improved the literature review. Notably, we have added one paragraph to clarify that we have filled at least theoretical gaps (please see the last paragraph of the section entitled “2. Literature Review and Hypotheses”). The revisions have been highlighted in red.

Reviewer #3: Method
The methodology section describes a methodical and transparent way to investigate the connections between the rookie STEM teachers' personality traits (PT), perceived social support (PSS), work engagement (WE), and job satisfaction (JS). It makes sense to utilise a quantitative cross-sectional survey, and structural equation modelling (SEM) is a suitable method for examining the intricate links suggested by the study's hypotheses. The study's dedication to ethical research procedures is demonstrated by the thorough explanation of the data collection procedure, which takes ethical considerations and the participation of instructional administration departments into account. Furthermore, a methodologically sound approach is demonstrated by the inclusion of control variables like age and gender and the use of AMOS software for SEM. However, there is still room for improvement. Although effective, the cross-sectional design restricts the capacity to establish causation between the variables under investigation. Longitudinal designs may prove useful in future studies to better capture changes over time and more clearly demonstrate causal linkages. Self-reported metrics run the risk of introducing bias. Further research with objective measures or third-party evaluations may improve the validity of the results.
-A more thorough description of the pilot testing procedure, including the justification for item removal and how the pilot test influenced the final survey instrument, would be beneficial to include in the technique section.

Authors’ responses: Thank you so much for your favorable comment.

Firstly, we have followed your suggestions and included more details regarding how future studies can improve our methodology in the subsection entitled “5.3 Limitations and future research”. The revisions have been highlighted in red.

Secondly, we have followed your suggestions and included more details regarding our pilot test. The revisions have been highlighted in red.

Reviewer #3: Additional information
- Theoretical Contribution: Although the study adds to our knowledge of novice STEM teachers' job satisfaction, it might incorporate additional theories or frameworks to enhance the analysis beyond the JD-R model. Examining other ideas might provide more in-depth understanding of the intricate interactions between various elements influencing work satisfaction.

Authors’ responses: Thank you for this comment. We have followed your suggestions and included more details regarding how future studies can incorporate additional theories or frameworks to enhance the analysis beyond the JD-R model in the subsection entitled “5.3 Limitations and future research”. The revisions have been highlighted in red.

Reviewer #3: - Practical Implications: While useful, the practice recommendations are a little too general. More targeted, doable solutions that are adapted to the particular difficulties faced by the educational sector will improve the piece. Giving specific instances of effective programmes or interventions could make these recommendations more useful.

Authors’ responses: Thank you for this comment. We have followed your advice and included more details in the subsection entitled “5.2 Practical Implications”. The revisions have been highlighted in red.

Reviewer #3: - Limitations and Future Research: It is appropriate to acknowledge the limitations of this study, including its cross-sectional design and reliance on self-reported data. The conversation could be broadened, though, in order to address potential biases in greater detail and offer particular methodological enhancements, such mixed-methods or longitudinal designs, for upcoming research. Innovativeness: Although the study offers insightful conclusions, it could be more innovative to investigate lesser-known or recently discovered factors that influence job satisfaction, such as the effects of global crises like the COVID-19 pandemic, policy changes in education, or technological developments in the classroom.

Authors’ responses: Thank you for this comment. Firstly, we have followed your advice and highlighted the methodology limitations. Secondly, we have followed your advice and suggested that future studies may investigate lesser-known or recently discovered factors that influence job satisfaction in the subsection entitled “5.3 Limitations and future research”. The revisions have been highlighted in red.

Round 2

Reviewer 2 Report

Comments and Suggestions for Authors

The manuscript presents a compelling investigation into the antecedents of job satisfaction among novice STEM teachers, a subject of significant importance in educational research. The integration of personality traits, perceived social support, and work engagement into the proposed model is particularly noteworthy. I’m pleased with the authors revisions

Reviewer 3 Report

Comments and Suggestions for Authors

Congrats on your work